# Two Sides of the Same Optimization Coin: Model Degradation and Representation Collapse in Graph Foundation Models

## Abstract

Graph foundation models (GFMs), inspired by the success of LLMs, are designed to learn the optimal embedding function from multi-domain text-attributed graphs (pre-training) for the downstream cross-task generalization capability (fine-tuning). During our investigation, graph vector quantized-masked autoencoder (gVQ-MAE) stands out among the increasingly diverse landscape of GFM architectures. This is attributed to its ability to jointly encode topology and textual attributes from multiple domains into discrete embedding spaces with clear semantic boundaries. Despite its potential, domain generalization conflicts cause imperceptible pitfalls. In this paper, we instantiate two of them, and they are just like two sides of the same GFM optimization coin - Side 1 Model Degradation: The encoder and codebook fail to capture the diversity of inputs (e.g., social networks and molecular graphs); Side 2 Representation Collapse: The hidden embedding and codebook vector fail to preserve semantic separability due to constraints from narrow representation subspaces. These two pitfalls (sides) collectively impair the decoder and generate the low-quality reconstructed supervision, causing the GFM optimization dilemma during pre-training (coin). Through empirical investigation, we attribute the above challenges to *Information Bottleneck* and *Regularization Deficit*. To address them, we propose MoT (Mixture-of-Tinkers) - ❶ **Information Tinker for Two Pitfalls**, which utilizes an edge-wise semantic fusion strategy and a mixture-of-codebooks with domain-aware routing to improve information capacity. ❷ **Regularization Tinker for Optimization Coin**, which utilizes two additional regularizations to further improve gradient supervision in our proposed Information Tinker. Notably, as a flexible architecture, MoT adheres to the scaling laws of GFM, offering a controllable model scale. Compared to SOTA baselines, experiments on 22 datasets across 6 domains demonstrate that MoT achieves significant improvements in supervised (1.4%), few-shot (3.1%), and zero-shot (3.3%) scenarios.

## 1 Introduction

In recent years, graph neural networks (GNNs) have revolutionized relational data modeling by capturing structural inductive biases [7; 28; 10]. However, their reliance on domain- and task-specific design severely constrains generalization [8; 12], often requiring costly retraining for new scenarios. Recent advances in graph foundation models (GFMs) seek to leverage the self-supervised paradigm to extract semantic consensus (i.e., topology and textual attributes insights) from multi-domain text-attributed graphs during pre-training for better generalization in various graph downstream tasks.

*Why Graph-oriented GFMs and gVQ-MAEs?* Reviewing existing GFM frameworks, we provide a brief summary in Fig. 1. The taxonomy is ① Language-oriented methods [39; 11] convert graphs into flattened textual representations for the token encoder (e.g., transformer-based LLMs) and ② Graph-oriented methods [36; 27; 24] preserve text comprehension and structural integrity through dedicated architectures (e.g., the frozen LLM combined with trainable GNN). The key insights are ① Language-oriented GFMs irreversibly disrupt graphs, but graph-oriented GFMs employ tailored TAG processing and self-supervised paradigms to maintain topology-awareness; ② The available TAG pre-training corpora are tiny (GB), rendering the utility of the parameter-intensive transformers. Therefore, considering the GFM scaling law, the model scale of GNN is already sufficient (million).

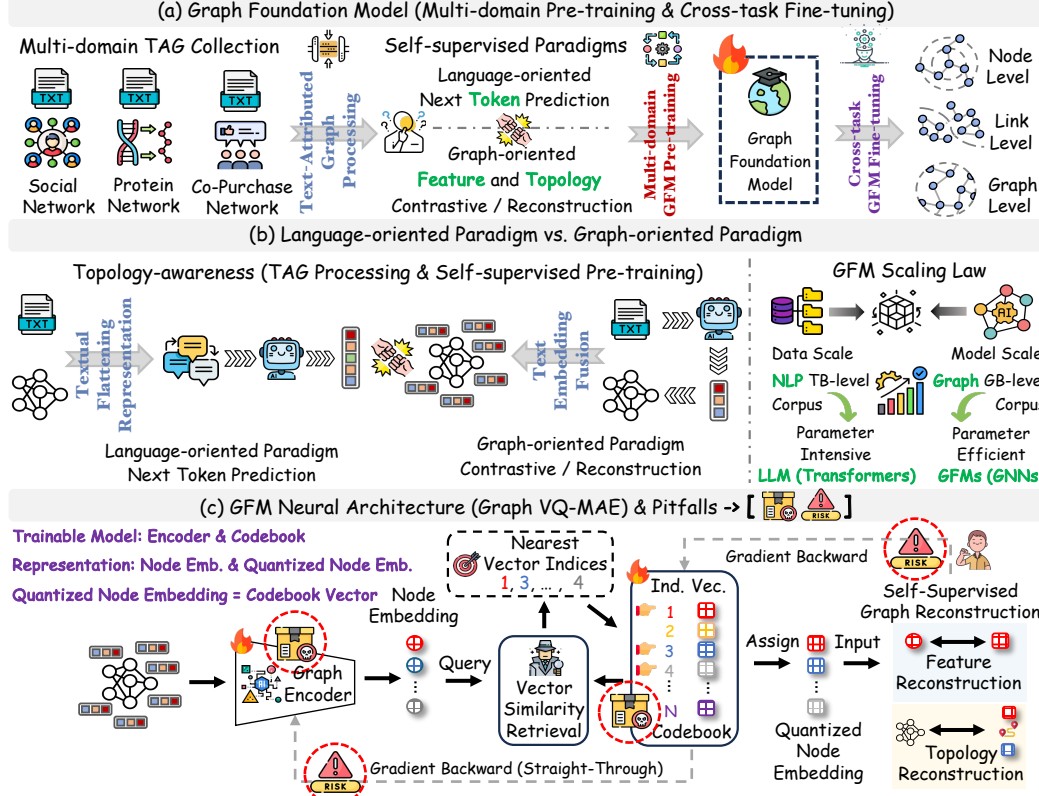

Figure 1: The overview of existing GFM studies and prevalent gVQ-MAE architecture.

Nevertheless, we emphasize the necessity of maintaining flexibility for GFM scale-up. Among the increasingly diverse landscape of graph-oriented GFMs, graph vector quantized-masked autoencoder (gVQ-MAE) [13; 18; 23] is promising, owing to ① Discrete embedding spaces enabled by vector quantization mitigate representation redundancy while preserving multi-domain graph separability [22]; ② Free-scaled encoder and codebook dynamically aligns model capacity toward flexibility and optimizes the memory-performance trade-off [15]. Please refer to Appendix A.1 for more details.

*What are gVQ-MAEs' Limitations and How to Solve Them?* Due to domain generalization conflicts, two underexplored yet interrelated pitfalls emerge in gVQ-MAEs during GFM pre-training. These pitfalls are just like two sides of the same optimization coin: Side 1 Model Degradation: The encoder and codebook often over-suppress domain-specific representations, especially for semantically conflicting inputs. Side 2 Representation Collapse: Progressive shrinkage of the latent space constrains hidden embeddings and codebook vectors to a narrow subspace [37]. This leads the decoder to over-utilize the limited embedding subset, generating low-quality reconstructed supervision and highlighting the optimization dilemma during pre-training (coin). In Sec. 3, we attribute these issues to the *Information Bottlenecks* and *Regularization Deficits*. To address them, we propose Mixture-of-Tinkers (MoT), which consists of: ❶ **Information Tinker for Two Pitfalls**, which utilizes an edge-wise semantic fusion strategy to enhance the encoder and employs a mixture-of-codebooks (MoC) with a tailored gated routing network. They jointly improve the information capacity of gVQ-MAE to maintain domain discriminability and representation diversity; ❷ **Regularization Tinker for Optimization Coin**, which utilizes the contrastive alignment and load-balancing constraint to improve Information Tinker further. They collaborate and serve as auxiliary gradient supervision.

**Our Contributions.** (1) *New Perspective.* We are the first to reveal the pitfalls of gVQ-MAEs in GFM optimization during pre-training and link them to *Information Bottlenecks* and *Regularization Deficits*. (2) *New Method.* We propose MoT for better optimization with a theoretical guarantee, which introduces edge-wise semantic fusion and MoC-enhanced vector quantization for two pitfalls, as well as two tailored regularizations for further improvements. (3) *SOTA Performance.* Extensive experiments demonstrate the superiority of MoT. In addition, by introducing MoC, we endow the model scale with flexible expansion, making it better suited for the GFM and showing great potential.

## 2 PRELIMINARIES

### 2.1 NOTATIONS AND PROBLEM FORMULATION

Consider a text-attributed graph (TAG) $\mathcal{G} = (\mathcal{V}, \mathcal{E}, \mathcal{T})$ with $|\mathcal{V}| = n$ nodes and $|\mathcal{E}| = m$ edges. $\mathcal{T}$ is the textual description for nodes and edges, and the adjacency matrix is $\mathbf{A}$. To achieve GFM, $\mathcal{T}$ is encoded into feature vectors $\mathcal{X}$ using a pre-trained text encoder. Now, we have $\mathcal{G}' = (\mathcal{V}, \mathcal{E}, \mathcal{X})$, which preserves both the topology and textual attributes. Based on this, GFM aims to learn an optimal embedding function $f_\theta$ over the multi-domain $\{\mathcal{G}'_i\}$ using self-supervised paradigms [9; 6; 16; 29]. For cross-task requirements, $f_\theta$ is fine-tuned with task-specific heads for better performance.

### 2.2 GRAPH FOUNDATION MODELS

**Language-oriented GFMs (✘).** They convert graphs into text sequences that encode nodes and edges using carefully designed syntactic rules, enabling the direct application of LLMs for graph understanding [4]. Specifically, during pre-training, they update the trainable embedding function (LLMs) using NLP optimization objectives, such as next-token prediction [8]. Despite inheriting key intuitions from LLMs, they suffer from irreversible topology disturbance and scalability concerns.

**Graph-oriented GFMs (✔).** They preserve text comprehension and structural integrity through dedicated architectures. Specifically, they typically employ frozen LLMs combined with trainable GNNs as the embedding function, enabling effective collaboration between topology and textual attributes [34; 14]. Based on this, during pre-training, they integrate reconstruction or contrastive self-supervised tasks, enabling the model to capture multi-domain TAG semantic consensus [3; 25].

### 2.3 GRAPH VECTOR QUANTIZED-MASKED AUTOENCODER

Most recent graph-oriented GFMs adopt gVQ-MAEs as the trainable module [32]. This architecture enables the joint encoding of topology and textual attributes into a discrete embedding space [38]. ① $\mathcal{G}' = (\mathcal{V}, \mathcal{E}, \mathcal{X}) \rightarrow$ ***Encoder*** $\rightarrow$ *Hid. Emb.*: To ensure generality, we use an encoder instantiated as any reasonable GNN capable of incorporating both node and edge features to generate $z$. ② *Hid. Emb.* $z \rightarrow$ ***Codebook*** $\rightarrow$ *Quan. Emb.*: To establish clear semantic boundaries, codebook $\mathcal{C}$ transforms continuous $z$ into discrete codebook vectors $e$ via similarity retrieval-based vector quantization:

$$e_j \rightarrow z_q, \ j = \arg\min_{e_i \in \mathcal{C}} \|z - e_i\|_2, \ \mathcal{C} = \{e_1, e_2, \ldots, e_K\}, \ e \in \mathbb{R}^d, \ z_q \in \mathbb{R}^d. \tag{1}$$

③ *Quan. Emb.* $z_q \rightarrow$ ***Decoder*** $\rightarrow \mathcal{G}'_r = (\mathcal{V}, \mathcal{E}_r, \mathcal{X}_r)$: To enable self-supervised training, gVQ-MAEs follow an autoencoder framework, where gradients are computed by the discrepancy between the reconstructed supervision $\mathcal{G}'_r$ and the original input $\mathcal{G}'$. To construct end-to-end gradient flow, the straight-through estimator (STE) [2; 35] is used to pass the non-differentiable quantization step.

In gVQ-MAEs, the Side 1 Model Degradation comprises a trainable encoder and codebook. The Side 2 Representation Collapse comprises the hidden embeddings and the codebook vectors. Besides, the optimization coin captures the overall GFM pre-training convergence, highlighting the role of gradient-based supervision. Our mask mechanisms and decoder are introduced in Appendix D.2.

## 3 EMPIRICAL INVESTIGATION

To further illustrate the model degradation and representation collapse, we first investigate two pitfalls (sides) via the embedding landscape (Fig. 2(a)-(c)). Then, we present the convergence curves to illustrate the direct effects of the two pitfalls in optimization (coin) (Fig. 2(d)). Please refer to Appendix B for more details about the experimental setup and analysis.

### 3.1 TWO SIDES: MODEL DEGRADATION AND REPRESENTATION COLLAPSE

◆ ***Questions*** → ***Observations*** → ***Conclusions***. *Questions:* ① Can encoder and codebook preserve separability and diversity? ② Can decoder achieve high-quality reconstruction? *Observations:* ① The low value in Fig. 2(a) and bimodal distribution in Fig. 2(b) exhibit significant semantic entanglement and suppression; ② The remarkable mismatch of $\mathcal{G}'$ and $\mathcal{G}'_r$ in Fig. 2(c) reveals decoding distortion. *Conclusions:* S1 Model Degradation and S2 Representation Collapse exist and are deeply intertwined.

(a) KL Divergence of Hidden Emb.  (b) Codebook Landscape  (c) Recons. Sup. Landscape  (d) Convergence Validation

Figure 2: Empirical results. The accuracy is reported through real-time downstream evaluation.

☆ **Key Insight → Solution**: ① *Key Insight (Sec. 5.2 ①):* In most TAGs, Appendix C shows that edge descriptions are limited (e.g., "*These two items are co-purchased*" in E-com), and their embeddings are frozen. This impairs message passing and leads to the notorious over-smoothing issue, where homogenized representations hinder the encoder. *Solution: Edge-wise Semantic Fusion.* We propose an enhanced graph encoder with edge-attributed message passing to achieve collaborative update of nodes and edges, thereby ensuring domain separability via improved **Information Flow in encoder**. ② *Key Insight (Sec. 5.2 ②):* A single codebook fails to capture the diverse semantics in multi-domain inputs, leading to sub-optimal vector quantization. *Solution: Mixture-of-Codebooks.* We propose this module inspired by the MoE architecture, employing multiple domain-specific codebooks (experts) alongside a tailored routing mechanism that selects the most appropriate codebook for quantization. This design enhances representation diversity by extending **Information Resource in codebook**.

## 3.2 SAME COIN: OPTIMIZATION DILEMMA IN MULTI-DOMAIN GFM PRE-TRAINING

◆ **Questions → Observations → Conclusions.** *Questions:* ① Can gVQ-MAEs stand out? ② Can Information Tinker improve gVQ-MAEs? *Observations:* Based on the curves in Fig. 2(d), we have ① Compared to other GFM architectures, gVQ-MAE demonstrates its superiority; ② Compared to other gVQ-MAEs, Information Tinker (Naive MoT) achieves the best performance but unsatisfactory convergence. *Conclusions:* Although Information Tinker is effective, it remains to be improved.

☆ **Key Insights → Solutions**: ① *Key Insight (Sec. 5.2 ③):* The conventional gVQ-MAE commitment loss fails to effectively optimize MoC, as it merely minimizes the pairwise distances between hidden embeddings and assigned codebook vectors, while neglecting the semantic conflicts among the codebooks. *Solution: Embedding-Vector Contrastive Alignment.* We pull hidden embeddings and assigned codebook vectors closer, while incorporating the repulsion to alleviate overcrowding among MoC, achieving **Adversarial Regularization in encoder and MoC**. ② *Key Insight (Sec. 5.2 ④):* The conventional MoE load loss fails to constrain MoC, as it only enforces average expert activation without accounting for the inter-codebook preferences. *Solution: MoC Load-balancing Constraint.* We dynamically redistribute MoC toward the domain-optimal load, while preventing individual codebooks from becoming high-density hubs, achieving **Domain-aware Regularization in MoC**.

## 4 MIXTURE-OF-TINKERS

In this section, we present the details of the MoT, and Fig. 3 illustrates its complete workflow.

### 4.1 INFORMATION TINKER

**Motivation.** Based on the empirical analysis in Sec. 3.1, the *Information Bottlenecks* between the encoder and codebook in gVQ-MAEs lead to S1 Model Degradation. To mitigate this, we design ① *Edge-wise Semantic Fusion* dynamically updates graph contextual information in the encoder to enhance **Information Flow**, and ② *Mixture-of-Codebooks* utilizes domain-specific codebooks separately to represent information from different semantic spaces to extend **Information Resource**.

**Edge-wise Semantic Fusion.** Existing GNNs for TAGs often naively integrate edge features by repeatedly aggregate them across layers [25; 12], which causes information redundancy when edges share identical features, mentioned in Sec. 3.1 and Appendix C. To resolve this, we propose Edge-wise Semantic Fusion, where each edge $\mathbf{e}_{uv}$ evolves by assimilating knowledge from its connected nodes $\mathbf{h}_u$ and $\mathbf{h}_v$. The detailed operation are described in Appendix D.1.

$$\mathbf{h}_u^{(l+1)} = \text{Agg}_1\left(\mathbf{h}_u^{(l)}; \text{Prop}_1\left\{\mathbf{h}_v^{(l)}, \mathbf{e}_{uv}^{(l)}\right\}\right), \quad \mathbf{e}_{uv}^{(l+1)} = \text{Agg}_2\left(\mathbf{e}_{uv}^{(l)}; \text{Prop}_2\left\{\mathbf{h}_u^{(l)}, \mathbf{h}_v^{(l)}\right\}\right). \quad (2)$$

Figure 3: The overview of vanilla gVQ-MAE and its enhancement by our proposed MoT.

**Mixture-of-Codebooks.** As previously noted, conventional single-codebook struggles to capture diverse cross-domain graph semantic patterns. While increasing codebook size naively expands representation capacity, it fails to fundamentally resolve the semantic conflicts inherent in the multi-domain GFM pre-training and further hinders optimization. In this context, we propose a sparsely activated Mixture-of-Codebooks $\{\mathcal{C}_1, \ldots, \mathcal{C}_M\}$, each specializing in distinct domains. Specifically, for an input node hidden embedding $\mathbf{h}_u$, we compute domain-specific activation scores via a gating network $G(\cdot)$ and select the Top-$k$ codebooks:

$$\mathcal{M}_{\text{active}} = \text{Top}_k\left(s_{u,1}, \ldots, s_{u,M}\right), \quad s_{u,i} = \text{MLP}_i\left(\mathbf{h}_u\right), \ \forall i \in \{1, \ldots, M\}. \tag{3}$$

The final quantized embedding $z_u$ combines outputs from active codebooks:

$$z_u = \sum_{m \in \mathcal{M}_{\text{active}}} \frac{s_{u,m}}{\sum_{j \in \mathcal{M}_{\text{active}}} s_{u,j}} \cdot \text{VQ}\left(\mathbf{h}_u, \mathcal{C}_m\right), \tag{4}$$

where $\text{VQ}(\cdot)$ denotes vector quantization to the nearest codebook, following the paradigm as Eq. (1).

This design achieves three critical properties: ① *Domain Specialization:* Each $\mathcal{C}_m$ auto-clusters semantically similar graph patterns; ② *Dynamic Capacity:* The capacity of MoC adaptively scales with the domain diversity and corpus scale of pre-training data, ensuring the optimal expressiveness; ③ *Gradient Stability:* Total codebook size scales as $O(M \cdot K)$ but only $O(k \cdot K)$ active units per sample. This normalized Top-$k$ weighting mitigates training instability in sparse routing.

## 4.2 REGULARIZATION TINKER

**Motivation.** As empirically demonstrated in Sec. 3.2, *Regularization Deficits* in conventional VQ lead to representation collapse in codebooks (i.e., the sub-optimal performance of naive MoT shown in Fig. 2(d)). To address this, we introduce two novel regularization objectives: ① *Embedding-Vector Contrastive Alignment* which minimizes the InfoNCE loss [16] for each node $u$ with pre-quantized embedding $\mathbf{h}_u$ and quantized code $z_u$ to achieve *Adversarial Regularization*; ② *Mixture-of-Codebooks Load-balancing Constraint* which aligns codebook usage with domain proportions, preventing codebook dominance and achieving *Domain-aware Regularization*.

**Embedding-Vector Contrastive Alignment.** Traditional gVQ-MAEs employ a commitment loss (e.g., MSE) to minimize the distance between embeddings $\mathbf{h}$ and quantized counterparts $z$ based on the nearest-neighbor retrieval. Despite its intuitiveness, this weakly constrained mechanism suffers from biased codebook learning, leading to amplified collapse in GFM pre-training. To address the dual challenges of codebook collapse and embedding collapse, we propose a Triple-Contrastive Loss that simultaneously achieves: ① *Alignment:* Attracts the corresponding $\mathbf{h}_i$-$z_i$ pairs via positive pairs. ② *Embedding Diversity:* Repels distinct hidden embeddings $\mathbf{h}_i$-$\mathbf{h}_j$ to mitigate embedding collapse. ③ *Codebook Dispersion:* Repels quantized codes $z_i$-$z_j$ to prevent token redundancy. Formally,

$$\mathcal{L}_{con} = -\frac{1}{n} \sum_{i=1}^{n} \log \frac{\exp\left(S\left(\mathbf{h}_i, z_i\right)/\tau\right)}{\sum_{j=1}^{n}\left(\exp\left(S\left(\mathbf{h}_i, z_j\right)/\tau\right) + \exp\left(S\left(\mathbf{h}_i, \mathbf{h}_j\right)/\tau\right) + \exp\left(S\left(z_i, z_j\right)/\tau\right)\right)}, \tag{5}$$

where $S(\cdot)$ computes cosine similarity, and $\tau$ is a temperature hyper-parameter.

**Mixture-of-Codebooks Load-balancing Constraint.** In MoC, a key challenge emerges when a small subset of codebooks dominates the gating mechanism, resulting in codebook collapse. In such cases, most inputs activate limited codebooks, significantly restricting the model's expressive capacity. This imbalance arises from the absence of explicit constraints to ensure equitable codebook utilization during training. To address this, we introduce a domain-aware load-balancing constraint, inspired by MoE but specifically designed for codebook specialization. This constraint encourages balanced usage by guiding inputs to preferentially activate their domain-specific codebooks:

$$\mathcal{L}_{load} = -\frac{1}{n} \sum_{i=1}^{n} \sum_{m=1}^{M} y_{i,m} \log \hat{y}_{i,m}, \quad \hat{y}_{i,m} = \frac{s_{i,m}}{\sum_{j=1}^{M} s_{i,j}}, \tag{6}$$

where $y_{i,m} \in \{0, 1\}$ denotes whether the node $i$ belongs the domain $m$. We ensure balanced corpus distribution across domains during pre-training, preventing skewed codebook activation.

### 4.3 THEORETICAL ANALYSIS

#### 4.3.1 WHY INFORMATION TINKER ALLEVIATES SEMANTIC ENTANGLEMENT?

**Definition 4.1. Information Bottleneck in GFMs.** Let $\mathcal{G}' = (\mathcal{V}, \mathcal{E}, \mathcal{X})$ denote an input graph with domain-specific semantics $S$. The Information Bottleneck principle aims to learn compressed representations $Z$ that maximize the **Mutual Information** of semantic relevance $I(Z; S)$ and $I(Z; \mathcal{E})$ while minimizing redundant information $I(Z; \mathcal{X})$. The optimal trade-off is governed by:

$$\min_{Z} \left[ I(Z; \mathcal{X}) - \alpha I(Z; S) - \beta I(Z; \mathcal{E}) \right], \quad \alpha > 0, \ \beta > 0. \tag{7}$$

Traditional gVQ-MAEs suffer from semantic entanglement due to ① *Static Edge Integration*: Naive aggregation schemes constrain edge-aware information flow, limiting $I(Z; \mathcal{E})$. ② *Single-Codebook Quantization*: A shared codebook $\mathcal{C}$ forces all domains into a single space, causing $I(Z; S) \leq \log K$ (bounded by codebook size). To break these limitations, we propose Information Tinker and present the following theoretical foundations to support its effectiveness (proofs are shown in Appendix E):

**Theorem 4.2. *Edge-wise Fusion Expands Information Flow.*** *Let $Z_{\text{vanilla}}$ and $Z_{\text{MoT}}$ denote node embeddings generated by a vanilla GNN and our Edge-wise Fusion (Eq. (2)), respectively. Then:*

$$I(Z_{\text{MoT}}; \mathcal{E}) \geq I(Z_{\text{vanilla}}; \mathcal{E}) + \gamma \sum_{\mathbf{e}_{uv} \in \mathcal{E}} \mathbb{E} \left[ \|\nabla_{\mathbf{e}_{uv}} \mathbf{h}_u\|^2 \right], \tag{8}$$

*where $\gamma = \frac{\alpha^2}{4}$, $\alpha$ is the Lipschitz constant of the activation function $\sigma$, and $\nabla_{\mathbf{e}_{uv}} \mathbf{h}_u$ is the gradient of node embedding $\mathbf{h}_u$ w.r.t. edge feature $\mathbf{e}_{uv}$.*

**Theorem 4.3. *Mixture-of-Codebooks Enhance Information Resource.*** *For $M$ domain-specific codebooks $\{\mathcal{C}_1, \ldots, \mathcal{C}_M\}$, each with $K$ vectors, the maximum semantic mutual information scales as:*

$$\max I(Z; S) \geq \log (M \cdot K). \tag{9}$$

*This strictly dominates the single-codebook upper bound $\max I(Z; S) \leq \log K$.*

#### 4.3.2 HOW CONTRASTIVE ALIGNMENT MITIGATES REPRESENTATION COLLAPSE?

**Definition 4.4. Representation Collapse.** Let $\mathbb{Z} \subseteq \mathbb{R}^d$ be the latent space of embedding $Z \in \mathbb{R}^d$. Representation collapse occurs when $\dim(\text{span}(\mathbb{Z})) \ll d$.

To combat collapse, our proposed Triple-Contrastive Loss $\mathcal{L}_{con}$ in Eq. (5) promotes: ① *Alignment* — minimizing the distance between positive pairs $(\mathbf{h}_i, z_i)$; ② *Uniformity* — maximizing the separation of negative pairs $(\mathbf{h}_i, \mathbf{h}_j)$ and $(z_i, z_j)$. Based on this, we leverage the hypersphere space to analyze their gradient-level optimization trajectories. It provides geometric intuition into how the optimization objective promotes alignment and uniformity and elucidates the underlying optimization dynamics that drive representation dispersion and prevent collapse, thereby enhancing interpretability.

**Theorem 4.5. *Contrastive Loss Induces Uniformity.*** *Minimizing the contrastive loss $\mathcal{L}_{con}$ approximates maximizing the pairwise angular distances:*

$$\min \mathcal{L}_{con} \propto \max \mathbb{E}_{\mathbf{h}_i, \mathbf{h}_j} \left[ \arccos \left( \frac{\mathbf{h}_i \cdot \mathbf{h}_j}{\|\mathbf{h}_i\| \cdot \|\mathbf{h}_j\|} \right) \right] \propto \max \mathbb{E}_{z_i, z_j} \left[ \arccos \left( \frac{z_i \cdot z_j}{\|z_i\| \cdot \|z_j\|} \right) \right]. \tag{10}$$

Table 1: Performance on fine-tuning setting. We report accuracy for node/edge-level tasks and AUC score for graph-level tasks. The best and sub-best results are marked in **Bold Red** and **Bold Blue**.

| Model | Node Classification | | | | Link Classification | | Graph Classification | |
|---|---|---|---|---|---|---|---|---|
| | Cora | WikiCS | Reddit | History | WN18RR | FB15K237 | HIV | MUV |
| GCN | $74.62_{\pm0.18}$ | $74.27_{\pm0.09}$ | $63.94_{\pm0.14}$ | $75.90_{\pm0.56}$ | $73.47_{\pm0.06}$ | $78.65_{\pm0.12}$ | $62.89_{\pm0.46}$ | $56.72_{\pm1.03}$ |
| GraphMAE | $74.19_{\pm0.42}$ | $78.77_{\pm0.36}$ | $61.40_{\pm0.55}$ | $75.31_{\pm0.87}$ | $71.09_{\pm0.59}$ | $82.16_{\pm0.13}$ | $64.84_{\pm1.42}$ | $65.91_{\pm0.94}$ |
| GIANT | $76.04_{\pm0.47}$ | $79.82_{\pm0.23}$ | $61.94_{\pm0.29}$ | $77.89_{\pm0.65}$ | $82.80_{\pm0.33}$ | $81.44_{\pm0.19}$ | $61.16_{\pm1.87}$ | $62.05_{\pm1.33}$ |
| GFT | $77.14_{\pm1.73}$ | $77.76_{\pm0.60}$ | $76.73_{\pm0.81}$ | **$84.12_{\pm0.62}$** | $94.16_{\pm0.21}$ | $86.84_{\pm0.61}$ | $70.29_{\pm2.48}$ | $66.06_{\pm2.79}$ |
| OFA | $75.61_{\pm0.87}$ | $77.72_{\pm0.65}$ | $73.61_{\pm0.90}$ | $83.45_{\pm0.78}$ | **$97.22_{\pm0.18}$** | $95.77_{\pm0.01}$ | $71.89_{\pm2.15}$ | $70.81_{\pm1.47}$ |
| SAMGPT | $76.29_{\pm0.31}$ | $73.96_{\pm0.26}$ | $66.40_{\pm0.59}$ | $80.98_{\pm0.10}$ | $75.46_{\pm0.20}$ | $86.28_{\pm0.31}$ | $66.40_{\pm0.59}$ | $69.24_{\pm0.14}$ |
| UniGraph | $76.43_{\pm0.55}$ | $79.98_{\pm1.21}$ | $74.46_{\pm0.75}$ | $83.27_{\pm0.92}$ | $85.45_{\pm0.34}$ | $94.81_{\pm1.32}$ | $71.23_{\pm1.93}$ | $69.12_{\pm1.55}$ |
| MoT-st-tiny | $83.77_{\pm1.34}$ | $80.16_{\pm1.78}$ | **$78.47_{\pm1.65}$** | $79.54_{\pm0.85}$ | $91.04_{\pm1.15}$ | $92.15_{\pm1.25}$ | $71.86_{\pm2.14}$ | $68.33_{\pm1.87}$ |
| MoT-st-base | **$84.31_{\pm1.78}$** | **$82.98_{\pm1.31}$** | $78.03_{\pm0.89}$ | $83.77_{\pm0.78}$ | **$94.62_{\pm0.21}$** | **$96.24_{\pm0.57}$** | **$72.89_{\pm2.04}$** | **$71.52_{\pm1.23}$** |
| MoT-st-large | **$85.05_{\pm0.51}$** | **$82.94_{\pm1.97}$** | **$78.05_{\pm1.47}$** | **$84.13_{\pm0.72}$** | $94.01_{\pm0.38}$ | **$96.88_{\pm0.42}$** | **$73.45_{\pm1.96}$** | **$71.18_{\pm1.45}$** |

Table 2: Performance on few-shot setting. We report accuracy for node/edge-level tasks and AUC score for graph-level tasks. The best and sub-best results are marked in **Bold Red** and **Bold Blue**.

| Model | Cora - 5way | | | | History - 5way | | | |
|---|---|---|---|---|---|---|---|---|
| | 10-shot | 5-shot | 3-shot | 0-shot | 10-shot | 5-shot | 3-shot | 0-shot |
| GraphMAE | $65.24_{\pm6.87}$ | $64.33_{\pm7.12}$ | $60.18_{\pm8.05}$ | $51.47_{\pm9.14}$ | $54.89_{\pm7.33}$ | $53.62_{\pm8.78}$ | $48.24_{\pm9.15}$ | $39.18_{\pm8.25}$ |
| GIANT | $65.05_{\pm7.14}$ | $63.91_{\pm8.22}$ | $62.33_{\pm9.08}$ | $54.62_{\pm7.01}$ | $56.33_{\pm6.95}$ | $51.24_{\pm7.87}$ | $50.86_{\pm8.44}$ | $38.33_{\pm9.12}$ |
| GFT | $69.33_{\pm8.62}$ | $68.67_{\pm9.91}$ | $64.00_{\pm9.05}$ | $61.04_{\pm7.64}$ | $61.33_{\pm8.84}$ | $60.04_{\pm9.16}$ | $59.33_{\pm7.77}$ | $44.67_{\pm6.53}$ |
| OFA | $70.15_{\pm7.24}$ | $67.33_{\pm8.85}$ | $65.24_{\pm9.96}$ | $59.18_{\pm8.45}$ | $60.45_{\pm8.15}$ | $58.78_{\pm7.89}$ | $56.24_{\pm8.02}$ | $43.87_{\pm7.78}$ |
| SAMGPT | $67.42_{\pm8.15}$ | $65.33_{\pm9.04}$ | $65.18_{\pm9.12}$ | $58.89_{\pm9.45}$ | $61.15_{\pm7.78}$ | $59.24_{\pm8.15}$ | $57.33_{\pm8.89}$ | $45.62_{\pm8.04}$ |
| UniGraph | $74.43_{\pm8.55}$ | $73.98_{\pm7.21}$ | $73.46_{\pm7.75}$ | $65.27_{\pm6.92}$ | **$65.45_{\pm4.34}$** | $61.81_{\pm8.32}$ | $58.23_{\pm7.93}$ | $44.12_{\pm6.55}$ |
| MoT-st-tiny | **$80.53_{\pm5.85}$** | **$79.37_{\pm5.50}$** | **$77.60_{\pm5.71}$** | $67.07_{\pm7.46}$ | $63.47_{\pm6.78}$ | $60.47_{\pm4.14}$ | $59.07_{\pm3.34}$ | $45.87_{\pm4.72}$ |
| MoT-st-base | **$80.93_{\pm4.51}$** | $78.67_{\pm4.87}$ | $74.73_{\pm4.77}$ | **$68.73_{\pm5.63}$** | **$65.68_{\pm5.28}$** | **$64.60_{\pm4.27}$** | **$62.93_{\pm3.17}$** | **$46.53_{\pm5.15}$** |
| MoT-st-large | **$82.27_{\pm3.41}$** | **$80.80_{\pm2.89}$** | **$79.47_{\pm3.55}$** | **$68.40_{\pm6.26}$** | $65.24_{\pm4.95}$ | **$64.95_{\pm4.05}$** | **$63.86_{\pm3.45}$** | **$46.87_{\pm4.92}$** |
| Model | WN18RR - 5way | | | | HIV - 2way | | | |
| | 10-shot | 5-shot | 3-shot | 0-shot | 10-shot | 5-shot | 3-shot | 0-shot |
| GraphMAE | $67.15_{\pm7.78}$ | $65.24_{\pm8.15}$ | $62.33_{\pm8.89}$ | $45.47_{\pm9.24}$ | $52.84_{\pm6.87}$ | $52.15_{\pm7.45}$ | $52.24_{\pm8.02}$ | $50.33_{\pm8.15}$ |
| GIANT | $66.86_{\pm6.98}$ | $65.19_{\pm7.78}$ | $63.95_{\pm8.45}$ | $48.79_{\pm8.89}$ | $51.16_{\pm5.95}$ | $51.86_{\pm6.24}$ | $51.15_{\pm7.01}$ | $50.45_{\pm7.78}$ |
| GFT | $73.02_{\pm9.43}$ | $71.33_{\pm7.98}$ | $70.67_{\pm8.11}$ | $50.00_{\pm9.93}$ | $57.75_{\pm9.45}$ | **$57.78_{\pm8.12}$** | $55.06_{\pm9.43}$ | $52.10_{\pm7.76}$ |
| OFA | $72.24_{\pm8.15}$ | $70.86_{\pm8.89}$ | $68.24_{\pm9.45}$ | $51.33_{\pm4.12}$ | $55.89_{\pm7.78}$ | $55.24_{\pm8.15}$ | $54.86_{\pm8.89}$ | $51.24_{\pm9.45}$ |
| SAMGPT | $69.89_{\pm9.24}$ | $68.15_{\pm9.97}$ | $65.24_{\pm7.45}$ | $48.23_{\pm6.15}$ | $56.15_{\pm8.89}$ | $54.24_{\pm9.45}$ | $53.33_{\pm8.12}$ | $51.86_{\pm7.78}$ |
| UniGraph | **$76.43_{\pm5.55}$** | $74.98_{\pm4.21}$ | $72.46_{\pm7.75}$ | $52.27_{\pm6.92}$ | $55.45_{\pm5.34}$ | $54.81_{\pm4.32}$ | $54.23_{\pm7.93}$ | $51.12_{\pm8.55}$ |
| MoT-st-tiny | $75.87_{\pm5.29}$ | $73.33_{\pm5.78}$ | $72.80_{\pm5.29}$ | $50.73_{\pm2.52}$ | $57.86_{\pm5.25}$ | $56.47_{\pm5.78}$ | $56.07_{\pm6.14}$ | $52.87_{\pm7.25}$ |
| MoT-st-base | $76.27_{\pm3.64}$ | **$76.40_{\pm3.97}$** | **$74.80_{\pm4.18}$** | **$52.93_{\pm7.39}$** | **$58.80_{\pm5.69}$** | $57.60_{\pm5.24}$ | **$56.40_{\pm5.87}$** | **$53.53_{\pm6.78}$** |
| MoT-st-large | **$78.24_{\pm4.87}$** | **$75.95_{\pm5.24}$** | **$75.15_{\pm5.78}$** | **$53.24_{\pm6.89}$** | **$58.45_{\pm5.12}$** | **$58.86_{\pm5.45}$** | **$56.24_{\pm5.98}$** | **$53.87_{\pm6.45}$** |

## 5 EXPERIMENTS

To validate the superiority of MoT, we conduct comprehensive experiments. We aim to answer: **Q1**: Does MoT outperform SOTA baselines in supervised/few-shot/zero-shot scenarios while adhering to GFM scaling laws? **Q2**: How do Information and Regularization Tinker alleviate model degradation and collapse? **Q3**: Is MoT resilient to data-scale variations and hyper-parameter sensitivity? **Q4**: Does MoT achieve practical time-accuracy trade-offs? The implementation details and MoT variants are introduced in Appendix F and A.2. Additional experimental results and hyper-parameter settings can be found in Appendix G and H. Unless otherwise specified, MoT refers to the MoT-st-base.

### 5.1 OVERALL PERFORMANCE

To answer **Q1**, we conduct systematic evaluations across three fundamental graph learning tasks (node/edge/graph classifications) under two distinct learning paradigms: (1) supervised fine-tuning and (2) few-/zero-shot transfer. We compare MoT with three categories of baselines: supervised GNN (GCN), unsupervised GNNs (GraphMAE, GIANT), and GFMs (GFT, OFA, SAMGPT, UniGraph).

Table 1 demonstrates MoT variants' strong performance in supervised scenarios, where even the lightest variant MoT-st-tiny exceeds all baselines. Table 2 highlights MoT's adaptability in data-scarce scenarios, which consistently outperforms existing methods across few-/zero-shot learning settings.

Table 3: Ablation on two tinkers for GFM pitfalls and optimization coin.

| Model | Cora | WikiCS | Reddit | History | WN18RR | HIV |
|---|---|---|---|---|---|---|
| *Information Tinker* | | | | | | |
| w/o. Fusion | $81.05_{\pm2.31}$ | $78.10_{\pm1.89}$ | $76.91_{\pm1.77}$ | $80.33_{\pm0.98}$ | $89.40_{\pm0.45}$ | $72.22_{\pm3.50}$ |
| w/o. MoC | $83.77_{\pm3.34}$ | $80.16_{\pm1.78}$ | $78.03_{\pm1.65}$ | $79.54_{\pm0.85}$ | $91.04_{\pm3.47}$ | $71.86_{\pm2.14}$ |
| *Regularization Tinker* | | | | | | |
| w/o. $\mathcal{L}_{con}$ | $82.11_{\pm4.02}$ | $78.90_{\pm3.11}$ | $74.20_{\pm2.05}$ | $77.22_{\pm1.89}$ | $90.11_{\pm1.22}$ | $69.90_{\pm3.78}$ |
| w/o. $\mathcal{L}_{load}$ | $83.12_{\pm2.89}$ | $78.33_{\pm1.78}$ | $75.11_{\pm1.45}$ | $77.12_{\pm0.89}$ | $92.91_{\pm0.78}$ | $68.89_{\pm2.67}$ |
| **MoT** | $\mathbf{84.31}_{\pm1.78}$ | $\mathbf{82.98}_{\pm1.31}$ | $\mathbf{78.47}_{\pm0.89}$ | $\mathbf{83.77}_{\pm0.78}$ | $\mathbf{94.62}_{\pm0.57}$ | $\mathbf{72.89}_{\pm2.04}$ |

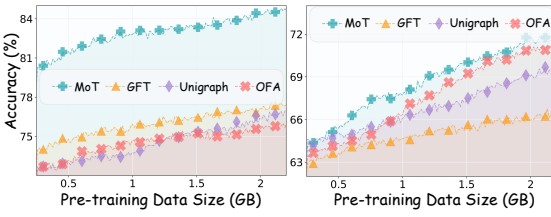

Figure 4: Perform. on Cora (left) and HIV (right).

Table 4: Impact on pre-training datasets.

| Datasets | WikiCS | FB15K237 | HIV |
|---|---|---|---|
| Target | $80.16_{\pm1.78}$ | $93.88_{\pm0.42}$ | $69.29_{\pm2.48}$ |
| Remaining | $81.44_{\pm1.22}$ | $94.92_{\pm0.38}$ | $72.03_{\pm2.15}$ |
| Tar. Dom. | $80.98_{\pm1.21}$ | $94.81_{\pm1.32}$ | $71.23_{\pm1.93}$ |
| Rem. Dom. | $79.33_{\pm1.45}$ | $94.24_{\pm0.57}$ | $71.16_{\pm2.04}$ |
| **All** | $\mathbf{82.98}_{\pm1.31}$ | $\mathbf{96.24}_{\pm0.57}$ | $\mathbf{72.89}_{\pm2.04}$ |

Compared to traditional GNNs, MoT benefits from large-scale pre-training. Compared to existing GFMs, MoT achieves superior generalization through its dual-tinker architecture: ① the Information Tinker dynamically fuses edge semantics to prevent model degradation, and ② the Regularization Tinker enforces geometric constraints via contrastive alignment to avert representation collapse. Such design achieves enhanced generalization by universalizing structural patterns across domains.

Crucially, MoT demonstrates remarkable adherence to GFM scaling laws across all evaluation dimensions. As model scale increases from tiny to large variants, we observe consistent performance improvements while maintaining stable variance patterns. MoT-st-large achieves peak performance on 6 of 8 fine-tuning tasks and 15 of 16 few-shot settings, though notably underperforms MoT-st-base in certain cases. While MoT variants successfully follow the scaling law, several critical issues remain unaddressed. For instance, insufficient pre-training data may lead to suboptimal parameter utilization and diminished performance returns, highlighting substantial room for future research.

## 5.2 ABLATION STUDY

To address **Q2**, we conduct ablation studies isolating core components of MoT, as shown in Table 3. We evaluate four critical variations by disabling Information Tinker (w/o. Fusion and w/o. MoC) and Regularization Tinker (w/o. $\mathcal{L}_{con}$ and w/o. $\mathcal{L}_{load}$). ① *Edge Semantic Fusion Ablation.* We replace the edge-wise semantic fusion with a naive message-passing, which causes significant performance degradation across all domains and tasks, confirming that our dynamic edge feature integration is essential for addressing model degradation. ② *MoC Ablation.* We replace the MoC with a single codebook, which causes performance collapse, confirming its critical role in preventing representation collapse. MoC preserves domain semantics and captures transferable domain invariances, which are essential for cross-domain generalization. ③ *Contrastive Loss Ablation.* We substitute the $\mathcal{L}_{con}$ with standard commitment loss [22] for codebook updates, which results in catastrophic performance degradation. By enforcing geometric separability among cross-domain representations, $\mathcal{L}_{con}$ prevents representation collapse in codebook and embeddings. ④ *Load Balancing Ablation.* We disable $\mathcal{L}_{load}$ and use the traditional load loss [19], leading to severe routing imbalance. This proves the constraint's necessity for balanced resource allocation in MoC, mitigating expert specialization bias.

## 5.3 ROBUSTNESS ANALYSIS

To answer **Q3**, we validate MoT's stability under different pre-training datasets and hyper-parameters.

**Data Scaling Law.** We investigate the impact of pre-training data scale on model performance. Fig. 4 reveals a positive correlation between pre-training data scale and performance, where MoT consistently outperforms baselines across all data scales. Crucially, Table 4 reveals that performance exhibits no significant dependence on whether the target dataset or domain is included during pre-training, showing MoT's exceptional transfer learning capabilities and domain-agnostic generalization.

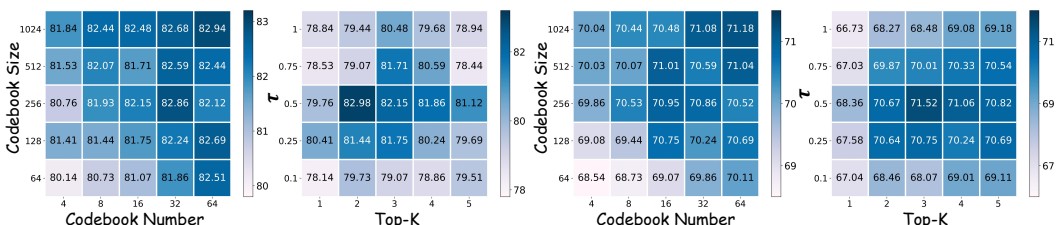

Figure 5: Sensitivity analysis on WikiCS (left two) and MUV (right two).

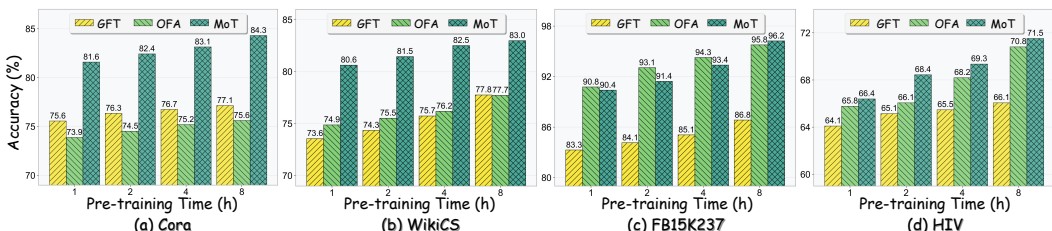

Figure 6: Pre-training efficiency comparison across multiple datasets and tasks.

**Hyper-parameter Sensitivity.** We systematically analyze the impact of three key hyper-parameters: the scale of the codebook, the Top-$k$ value in the MoC, and the temperature coefficient $\tau$ in contrastive learning. As demonstrated in Fig. 5, MoT maintains robust performance across all configurations. Regarding the codebook structure, performance improves as both the codebook size and the number of codebooks increase. However, larger codebooks also introduce higher computational and memory costs. The routing mechanism and contrastive learning also play critical roles in model behavior. Excessively large or small Top-$k$ and $\tau$ can lead to performance degradation. We suggest that in the experiment, the Top-$k$ be set to 2 or 3, the $\tau$ be set to around 0.5, the size of the codebook be set to 16 codebooks of size 256, which achieves an optimal balance between performance and efficiency.

## 5.4 EFFICIENCY ANALYSIS

To answer **Q4**, we evaluate MoT's computational efficiency and report the real-time downstream evaluation. As shown in Fig. 6, MoT achieves superior performance with significantly reduced pre-training time compared to existing methods. The dual-tinker architecture enables this efficiency through two key mechanisms: (1) the mixture-of-codebooks reduces redundant computations by activating domain-specific experts dynamically, and (2) the regularization tinker maintains stable convergence without expensive hyper-parameter tuning. MoT achieves higher performance even with shorter pre-training time, indicating faster convergence of our method.

## 6 CONCLUSION

In this paper, we identify critical optimization dilemmas in GFMs, manifested as model degradation and representation collapse. To address this, we proposed MoT, a novel framework that integrates an Information Tinker with edge-wise semantic fusion and mixture-of-codebooks, and a Regularization Tinker with contrastive alignment and load-balancing constraints. Theoretically, MoT provably expands information flow and mitigates collapse, as demonstrated by SOTA performances in extensive experiments across diverse datasets. However, our work has limitations: (1) The scale and diversity of existing pre-training datasets remain limited and constrain the performance upper bound of MoT, particularly for large-scale variants. (2) MoT involves multiple hyperparameters that require careful manual tuning to achieve optimal performance, adding experimentation overhead. To overcome these, future work will focus on: (1) Promoting the development of larger, higher-quality TAGs to unlock fuller model potential. (2) Designing more adaptive mechanisms (e.g., self-adjusting routing networks) to reduce manual tuning costs and enhance robustness. We will also extend MoT to broader applications, including cross-modal alignment with LLMs. Our work establishes a flexible foundation for graph pre-training, and these efforts will further strengthen its practicality and generalization.

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

Table 5: The statistician of the GFMs. #Num. denotes the number of datasets, #Dom. is the number of domains, #Size is the dataset scale in GB, and gVQ-MAE* is the simplified version of gVQ-MAE.

| Model | #Param. | Pre-training Datasets | | | Target | Self-Supervised Signal | Architecture |
|---|---|---|---|---|---|---|---|
| | | #Num. | #Dom. | #Size | | | |
| GOFA [8] | 10M | 2 | 2 | 35 | Language | Graph & Language Tasks | Tailored GNN |
| UniGraph [4] | 180M | 15 | 5 | 40 | Language | Graph Reconstruction | gVQ-MAE* |
| AnyGraph [30] | 17M | 15 | 4 | 5 | Graph | Link Prediction | gVQ-MAE* |
| GFSE [3] | 20M | 12 | 5 | 5 | Graph | Graph Contrastive Learning | Tailored GNN |
| GIT [26] | 4M | 8 | 4 | 1 | Graph | Graph Reconstruction | gVQ-MAE* |
| OpenGraph [31] | 40M | 11 | 3 | 5 | Graph | Supervised Learning | Tailored GNN |
| OFA [12] | 29M | 8 | 3 | 1 | Graph | Supervised Learning | Tailored GNN |
| GQT [23] | 5M | 20 | 8 | 5 | Graph | Graph Reconstruction | gVQ-MAE |
| GFT [25] | 7M | 9 | 4 | 1 | Graph | Graph Reconstruction | gVQ-MAE |
| GraphCLIP [40] | 150M | 5 | 3 | 1 | Graph | Graph Contrastive Learning | Tailored GNN |
| RiemannGFM [20] | 40K | 6 | 3 | 1 | Graph | Graph Contrastive Learning | Tailored GNN |
| SAMGPT [33] | 280K | 7 | 4 | 1 | Graph | Graph Contrastive Learning | Tailored GNN |
| UniGraph2 [5] | 30M | 14 | 5 | 40 | Graph | Graph Reconstruction | gVQ-MAE* |
| **MoT-st-tiny** | **5M** | **22** | **6** | **2** | **Graph** | **Graph Reconstruction** | **gVQ-MAE** |
| **MoT-st-base** | **10M** | **22** | **6** | **2** | **Graph** | **Graph Reconstruction** | **gVQ-MAE** |
| **MoT-st-large** | **60M** | **22** | **6** | **2** | **Graph** | **Graph Reconstruction** | **gVQ-MAE** |
| **MoT-llama7b-tiny** | **100M** | **22** | **6** | **2** | **Graph** | **Graph Reconstruction** | **gVQ-MAE** |
| **MoT-llama7b-base** | **170M** | **22** | **6** | **2** | **Graph** | **Graph Reconstruction** | **gVQ-MAE** |
| **MoT-llama7b-large** | **450M** | **22** | **6** | **2** | **Graph** | **Graph Reconstruction** | **gVQ-MAE** |

# A STATISTICIAN AND DISCUSSIONS OF GRAPH FOUNDATION MODELS

## A.1 DISCUSSION OF EXISTING GFMs

We summarize key characteristics of most existing GFMs as shown in Table 5. This systematic comparison reveals fundamental divergences between language-oriented and graph-oriented GFMs. For example, language-oriented approaches such as UniGraph require significantly higher computational resources, with 180 million parameters and 40GB of pre-training data, whereas graph-oriented GFMs like RiemannGFM attain comparable performance using only 40 thousand parameters and 1GB of data. This disparity primarily arises from the differences in data and model paradigms. Specifically, language-oriented approaches rely heavily on text-based corpora, requiring parameter-intensive transformer variants to capture complex patterns embedded within flattened, topology-infused token sequences. In contrast, graph-oriented methods store textual information in vectorized form, significantly reducing storage requirements. Moreover, their explicit utilization of both feature and topology information enables the use of parameter-efficient GNNs to achieve strong self-supervised performance. This underscores the inherent efficiency advantages of graph-oriented GFMs.

Based on this, the widespread adoption of gVQ-MAE and its variants (denoted by *) across diverse domains underscores their effectiveness as a general-purpose framework for graph pre-training. This architectural preference is largely attributed to two fundamental advantages. First, the discrete embedding space introduced by vector quantization significantly reduces representational redundancy, which is especially beneficial in multi-domain graph pre-training where multiple inputs often exhibit semantic gap (topology and textual features). Second, the decoupled and dynamic design of the encoder and vector quantization codebook allows for flexible control over the trade-off between memory efficiency and model expressiveness. The encoder can be tailored to the complexity of individual domains, while the codebook can scale independently to accommodate the granularity of learned patterns, enabling effective pre-training on graphs of varying size, density, and semantics.

As for our proposed MoT, we utilize a substantially larger number of datasets-22 spanning 6 distinct domains-compared to prior methods such as GOFA, which employs only 2 datasets. Despite this breadth, our total pre-training data size (2GB) remains considerably smaller than that of most baselines. This discrepancy arises from fundamentally different dataset selection strategies. Specifically, while GOFA depends on a small number of large-scale datasets (e.g., MAG240M, 33GB), our framework intentionally emphasizes dataset diversity over size by integrating a wide array of smaller datasets to achieve comprehensive domain coverage. In our implementation, our weighted pre-training pipeline automatically subsamples excessively large graphs (e.g., using a 0.1× sampling rate for PCBA), ensuring balanced representation across domains.

## A.2 MoT Variant Specifications

The proposed MoT systematically investigates architectural scaling through six model variants, differentiated along two primary dimensions: text encoding methodology and vector quantization complexity. For textual feature extraction, we implement two distinct encoding pipelines: (1) a sentence transformer [17] generating 768-dimensional node and edge features, and (2) a frozen LLaMA-7B [21] model producing 4096-dimensional features. The former provides computationally efficient semantic encoding suitable for resource-constrained deployments, while the latter leverages large language model capabilities for capturing nuanced linguistic patterns at higher dimensionality. Due to higher computational demands without proportional performance improvements observed in MoT-llama variants under limited dataset scales, we focus experimental reporting on MoT-st.

We also develop three quantization architectures. The **tiny** variant employs a single codebook with 128 vectors, operating without gating mechanisms. This configuration serves with total capacity of 5M (st) or 100M (llama) parameters. Building upon this foundation, the **base** configuration introduces domain-aware processing through 6 dedicated codebooks, each maintaining 128 vectors with gated routing. This architecture expands representational capacity to 10M (st) or 170M (llama) parameters while enabling basic cross-domain adaptation. The **large** variant represents our most sophisticated quantization scheme, implementing 64 codebooks with 1024 vectors each. With total capacity reaching 60M (st) and 450M (llama) parameters, this configuration theoretically supports multi-granular encoding of structural motifs, semantic relations, and cross-domain patterns. The routing mechanism dynamically activates subsets of codebooks, where the gating network learns to distribute inputs across specialized quantization subspaces. While the **MoT-large** is theoretically capable of comprehensive multi-scale representation through its high-capacity codebooks, it currently faces implementation constraints due to insufficient training corpus scale. The architecture demonstrates remarkable scalability potential when future work addresses corpus scaling challenges. For MoT-tiny and MoT-large, our proposed Load-balancing Constraint ($\mathcal{L}_{load}$) is deactivated. In MoT-tiny, where the MoC module is omitted, this constraint becomes redundant. In MoT-large, we substitute $\mathcal{L}_{load}$ with conventional MoE balance loss to maintain experts importance equilibrium.

## B Detailed Implementation of Empirical Study

**KL Divergence of Hidden Embedding**. Fig. 2(a) visualizes embedding collapse through KL divergence metrics. To generate this heatmap, we first extract node embeddings from the final encoder layer of the pre-trained model after convergence. Node embeddings are grouped by their predefined domains, and the domain-wise mean embedding is computed for each category. Pairwise KL divergences between all domain embedding pairs are then calculated. Lower KL values (e.g., Bio-Web: 0.16) indicate severe distributional overlap, where hidden embeddings fail to distinguish domain-specific features. This pattern aligns with the hypothesis that existing GFMs struggle to preserve domain-specific semantics in hidden spaces and suffer from representation collapse.

**Codebook Landscape**. Fig. 2(b) analyzes the quantized embedding distribution to diagnose model degradation and representation collapse. We first generate node embeddings by encoding the pre-training dataset through their frozen encoders. These embeddings are then mapped to discrete latent codes via their codebooks, followed by PCA projection to 1D space for visualization. The blue density curve reveals a bimodal distribution (peaks at -0.2 and 0.8), where major of quantized vectors cluster within narrow ranges. It indicates severe representation collapse, as the model fails to utilize the latent space effectively, compressing diverse graph structures into repetitive patterns.

**Reconstructed Supervision Landscape**. Fig. 2(c) evaluates the fidelity of node feature reconstruction. After mapping quantized codebook embeddings to reconstructed features via the decoder, we compare their distributions against original node features (blue curve) through shared PCA transformation to ensure comparable latent space. The reconstructed node features (blue curve) peaks sharply at -0.4 with a narrow spread, while the blue curve follows a broad bimodal distribution (peaks at 0.2 and 0.75). This mismatch indicates severe reconstruction failure and the degradation severity.

**Convergence Validation**. Fig. 2(d) benchmarks the downstream task efficiency of GFMs by tracking real-time validation accuracy on the Cora in node classification task during pre-training. The outcome validates that MoT's architectural innovations mitigate model degradation and representation collapse, enabling efficient knowledge transfer to downstream tasks.

Table 6: The edge descriptions of experimental text-attributed graphs.

| Domain | Edge Description |
|---|---|
| Citation Network | Feature edge. Citation. |
| Wikipedia Page | Feature edge. Wikipedia page link. |
| Social Network | Feature edge. Connected users have replied to each other or are following relationships. |
| Knowledge Graph | Feature edge. Relation between two entities: *<relation name>*. |
| E-commerce | Feature edge. These two items are frequently co-purchased or co-viewed. |
| Molecular Network | Feature edge. Chemical bond. *<bond type>* bond, bond stereo is *<bond stereo>*, *(is/is not)* conjugated. |

## C    EDGE DESCRIPTION LIMITATIONS IN TEXT-ATTRIBUTED GRAPHS

We systematically catalog the raw edge descriptions of all text-attributed graphs used in Table 6. Our analysis reveals a pervasive limitation: *The edge texts exhibit extreme homogeneity, either through identical descriptions or rigid templates.* This uniformity severely constrains the informational value of edge features, as they fail to capture edge-specific semantic nuances. When processed by conventional GFMs using standard GNN encoders, these redundant edge descriptions contribute to over-smoothing phenomenon, where node representations become indistinguishable due to excessive homogenization of neighborhood information. This fundamental limitation motivates our proposed edge-wise semantic fusion strategy, which dynamically enriches edge representations by integrating contextual node information, breaking the representation collapse while preserving structural integrity.

## D    DETAILED IMPLEMENTATION OF PRE-TRAINING

### D.1    GRAPH ENCODER

We proceed with a detailed explanation of Eq. (2) to fully illustrate the operation of our edge-wise semantic fusion. This method propagates information by integrating both node and edge features, thereby enhancing graph information flow and effectively alleviating the representation collapse.

$$
\mathbf{h}_u^{(l+1)} = \sigma \left( \mathbf{W}_1^{(l)} \mathbf{h}_u^{(l)} + \frac{1}{|\mathcal{N}(u)|} \sum_{v \in \mathcal{N}(u)} \mathbf{W}_2^{(l)} \left( \mathbf{h}_v^{(l)} + \mathbf{e}_{uv}^{(l)} \right) \right),
$$

$$
\mathbf{e}_{uv}^{(l+1)} = \sigma \left( \mathbf{W}_3^{(l)} \mathbf{e}_{uv}^{(l)} + \frac{1}{2} \mathbf{W}_4^{(l)} \left( \mathbf{h}_u^{(l)} + \mathbf{h}_v^{(l)} \right) \right),
$$

$$(11)$$

where $\left\{ \mathbf{W}_i^{(l)} \right\}$, $i = 1, 2, 3, 4$ are learnable transformation matrices and $\sigma$ is the activation function.

### D.2    GRAPH RECONSTRUCTION

To achieve effective pre-training, self-supervised signals are essential. In our implementation, MoT employs dual masking strategies. Specifically, feature masking randomly obscures $p_f$ of dimensions in the $\mathcal{X}$, while topology masking removes $p_t$ of edges from the $\mathcal{E}$. These jointly generate a corrupted graph $\tilde{\mathcal{G}} = (\mathcal{V}, \mathcal{E} \odot \mathcal{M}_t, \mathcal{X} \odot \mathcal{M}_f)$, where $\odot$ denotes element-wise multiplication and $\mathcal{M}_f, \mathcal{M}_t$ are binary masking matrices. Based on this, the reconstruction process utilizes two specialized decoders to recover node features and graph topology from the quantized embeddings. Feature reconstruction is achieved by minimizing the Euclidean distance between the original and reconstructed features:

$$
\mathcal{L}_{feat} = \frac{1}{|\mathcal{V}|} \sum_{i \in \mathcal{V}} \left\| z_i^f - \mathbf{x}_i \right\|_2^2,
$$

$$(12)$$

where $z_i^f$ denotes the linearly projected node embeddings $z_i$ and $\mathbf{x}_i$ represents the original feature.

Topology reconstruction employs a negative sampling strategy to preserve graph connectivity patterns:

$$\mathcal{L}_{topo} = \sum_{(i,j)\in\mathcal{E}} -\frac{1}{|\mathcal{E}|} \log\left(\sigma\left(z_i^{t\top} z_j^t\right)\right) - \sum_{(i,j')\in\hat{\mathcal{E}}} \frac{1}{|\hat{\mathcal{E}}|} \log\left(1 - \sigma\left(z_i^{t\top} z_{j'}^t\right)\right), \tag{13}$$

with $\sigma(\cdot)$ as the sigmoid function that transforms pairwise embedding similarities into edge existence probabilities. $\mathcal{E}$ and $\hat{\mathcal{E}}$ denote the existing and non-existing edge sets, respectively. These components are unified through a weighted multi-task learning framework:

$$\mathcal{L} = \lambda_1 \mathcal{L}_{feat} + \lambda_2 \mathcal{L}_{topo} + \lambda_3 \mathcal{L}_{con} + \lambda_4 \mathcal{L}_{load}, \tag{14}$$

where hyper-parameters $\lambda_1$-$\lambda_4$ balance the contributions of different optimization objectives.

# E  PROOF OF THEORETICAL ANALYSIS

## E.1  PROOF OF THEOREM 4.2

Theorem 4.2 establishes a formal connection between edge-wise semantic fusion and information flow enhancement, providing a theoretical foundation for our architecture design in Sec. 4.1.

**Proof:** Using the variational lower bound for mutual information [1]:

$$I(Z;\mathcal{E}) \geq \mathbb{E}_{\mathcal{E},Z}\left[\log q(\mathcal{E}|Z)\right] + H(\mathcal{E}), \tag{15}$$

where $q(\mathcal{E}|Z)$ is a decoder reconstructing edge information.

Define $L_{\text{model}} = \mathbb{E}[\log q(\mathcal{E}|Z)]$. The mutual information gap is bounded by:

$$I(Z_{\text{MoT}};\mathcal{E}) - I(Z_{\text{vanilla}};\mathcal{E}) \geq L_{\text{MoT}} - L_{\text{vanilla}}. \tag{16}$$

Expand $L_{\text{MoT}}$ using the embedding increment $\Delta Z$ from edge fusion:

$$L_{\text{MoT}} = \mathbb{E}\left[\log q\left(\mathcal{E}|Z_{\text{vanilla}} + \Delta Z\right)\right], \quad \Delta Z = \frac{1}{|\mathcal{N}(u)|} \sum_{v\in\mathcal{N}(u)} \mathbf{W}_2 \mathbf{e}_{uv} + \mathcal{O}\left(\|\mathbf{e}\|^2\right). \tag{17}$$

Taylor expansion reveals the information gain mechanism:

$$L_{\text{MoT}} = L_{\text{vanilla}} + \mathbb{E}\left[\nabla_Z \log q \cdot \Delta Z\right] + \frac{1}{2}\mathbb{E}\left[\Delta Z^\top \nabla_Z^2 \log q \cdot \Delta Z\right]. \tag{18}$$

Replace the unsubstantiated inequality with a rigorous bound using the Lipschitz property of $\sigma$:

$$\mathbb{E}\left[\Delta Z^\top \nabla_Z^2 \log q \cdot \Delta Z\right] \geq \frac{1}{2}\mathbb{E}\left[\|\Delta Z\|^2\right], \quad \|\Delta Z\| \geq \alpha \|\nabla_{\mathbf{e}_{uv}} \mathbf{h}_u\|. \tag{19}$$

The causal structure of edge reconstruction ensures:

$$\mathbb{E}_{\mathbf{e}_{uv}}\left[\nabla_Z \log q \cdot \frac{\partial Z}{\partial \mathbf{e}_{uv}}\right] = \mathbb{E}_{\mathbf{e}_{uv}}\left[\frac{\partial \log q}{\partial \mathbf{e}_{uv}}\right] \geq 0. \tag{20}$$

This term quantifies the direct contribution of $\mathbf{e}_{uv}$ to reconstruction loss.

When edge features are independent of node embeddings, cross-terms vanish:

$$\mathbb{E}[\nabla_Z \log q \cdot \Delta Z] \geq 0. \tag{21}$$

Combining these effects yields the final bound:

$$\begin{aligned}
I(Z_{\text{MoT}};\mathcal{E}) - I(Z_{\text{vanilla}};\mathcal{E}) &\geq \mathbb{E}\left[\nabla_Z \log q \cdot \Delta Z\right] + \frac{1}{2}\mathbb{E}\left[\Delta Z^\top \nabla_Z^2 \log q \cdot \Delta Z\right] \\
&\geq \frac{1}{2}\mathbb{E}\left[\Delta Z^\top \nabla_Z^2 \log q \cdot \Delta Z\right] \\
&\geq \frac{1}{4}\mathbb{E}\left[\|\Delta Z\|^2\right] \\
&\geq \frac{\alpha^2}{4}\mathbb{E}\left[\|\nabla_{\mathbf{e}_{uv}} \mathbf{h}_u\|^2\right].
\end{aligned} \tag{22}$$

This proof establishes a direct information pathway ($\nabla_{\mathbf{e}_{uv}} \mathbf{h}_u$) that amplifies edge-aware signals, mathematically justifying why our edge-wise fusion outperforms traditional aggregation schemes.

### E.2 PROOF OF THEOREM 4.3

Theorem 4.3 quantifies the representational advantage of mixture-of-codebooks, explaining the multi-domain scalability.

**Proof:** Each domain $S_m$ is assigned a dedicated codebook $\mathcal{C}_m$. For codebook $\mathcal{C}_m$ and codeword $e_{m,i} \in \mathcal{C}_m$, the probability of correct domain-specific mapping is:

$$p(e_{m,i}, \mathcal{C}_m) = \frac{1}{K} \cdot \frac{1}{M}. \tag{23}$$

Assuming domain independence, joint entropy across $M$ codebooks:

$$H(S_1, \ldots, S_M) = -\log p(e_{m,i}, \mathcal{C}_m) = \log(M \cdot K). \tag{24}$$

Mutual information $I(Z; S)$ is bounded by:

$$I(Z; S) \geq H(S) - H(S|Z) = \log(M \cdot K) - \epsilon, \tag{25}$$

where $\epsilon \to 0$ under optimal routing.

This demonstrates how MoC overcomes the $\log K$ bottleneck of standard gVQ-MAEs. The $M \cdot K$ scaling explains why adding codebooks improves cross-domain generalization without increasing $K$.

In a nutshell, the above theorems systematically demonstrate from an information-theoretic perspective that MoT transcends the representational capacity limits of conventional methods, while enhancing the GFM semantic representation space under bounded quantization error.

### E.3 PROOF OF THEOREM 4.5

This lemma connects the contrastive loss geometry with collapse prevention.

**Proof:** The triple-contrastive loss in Eq. (5) is:

$$\mathcal{L}_{con} = -\frac{1}{n} \sum_{i=1}^{n} \log \frac{\exp(S(\mathbf{h}_i, z_i)/\tau)}{\sum_{j=1}^{n} (\exp(S(\mathbf{h}_i, z_j)/\tau) + \exp(S(\mathbf{h}_i, \mathbf{h}_j)/\tau) + \exp(S(z_i, z_j)/\tau))}.$$

As $\tau \to 0^+$, the dominant terms become:

$$\mathcal{L}_{con} \approx c_1 \mathbb{E}_{\mathbf{h}_i, \mathbf{h}_j} \left[ \|\mathbf{h}_i - \mathbf{h}_j\|^2 \right] + c_2 \mathbb{E}_{z_i, z_j} \left[ \|z_i - z_j\|^2 \right], \quad c_1, c_2 > 0. \tag{26}$$

Under hyperspherical constraint ($\|\mathbf{h}_i\| = 1, \|z_i\| = 1$):

$$\|\mathbf{h}_i - \mathbf{h}_j\|^2 = 2 - 2\cos\theta_{ij}, \quad \theta_{ij} = \arccos(\mathbf{h}_i \cdot \mathbf{h}_j). \tag{27}$$

Thus minimizing $\mathcal{L}_{con}$ is equivalent to maximizing angular distance $\theta_{ij}$.

We employ angular geometry to decode the contrastive loss dynamics, revealing how gradient forces naturally induce hyperspherical uniformity. This proves why our triple-contrastive design prevents the representation collapse common in gVQ-MAEs.

## F EXPERIMENTAL SETTING

### F.1 PRE-TRAINING DATASETS

Our pre-training corpus encompasses a diverse collection of 22 benchmark datasets spanning 6 distinct domains, as shown in Table 7. This multi-domain collection exhibits substantial variation in scale, ranging from small-scale academic networks (2,708 nodes in Cora) to massive e-commerce graphs (316K nodes with 19.3M edges in Products). To address the inherent imbalance in cross-domain graph dataset scales, we follow [12] and implement a sampling strategy that normalizes domain contributions during pre-training. The sampling weight can be found in the last column.

Table 7: The statistician of the pre-training datasets.

| Domain | Dataset | Avg. #Nodes | Avg. #Edges | #Graphs | Task | #Classes | #Weight |
|--------|---------|-------------|-------------|---------|------|----------|---------|
| Citation Network | Cora | 2,708 | 10,556 | 1 | Node | 7 | 10 |
| | CiteSeer | 3,186 | 8,450 | 1 | Node | 6 | 10 |
| | Pubmed | 19,717 | 44,324 | 1 | Node | 3 | 10 |
| | Arxiv | 169,343 | 2,315,598 | 1 | Node | 40 | 1 |
| Web Link | WikiCS | 11,701 | 431,726 | 1 | Node | 10 | 10 |
| Social Network | Reddit | 33,434 | 198,448 | 1 | Node | 2 | 10 |
| | Instagram | 11,339 | 144,010 | 1 | Node | 2 | 10 |
| Knowledge Graph | WN18RR | 40,943 | 93,003 | 1 | Link | 11 | 10 |
| | FB15K237 | 14,541 | 310,116 | 1 | Link | 237 | 10 |
| E-commerce | History | 41,551 | 358,574 | 1 | Node | 12 | 1 |
| | Computers | 87,229 | 721,081 | 1 | Node | 10 | 1 |
| | Photo | 48,362 | 500,939 | 1 | Node | 12 | 1 |
| | Sportsfit | 173,055 | 1,773,500 | 1 | Node | 13 | 1 |
| | Products | 316,513 | 19,337,745 | 1 | Node | 39 | 1 |
| Molecular Graph | BACE | 34.1 | 73.7 | 1,513 | Graph | 1 | 1 |
| | BBBP | 24.1 | 51.9 | 2,039 | Graph | 1 | 1 |
| | HIV | 25.5 | 54.9 | 41,127 | Graph | 1 | 1 |
| | PCBA | 25.9 | 56.1 | 437,929 | Graph | 128 | 0.1 |
| | MUV | 24.2 | 52.6 | 93,087 | Graph | 17 | 1 |
| | cyp450 | 24.5 | 53.0 | 16,896 | Graph | 5 | 1 |
| | toxcast | 18.8 | 38.5 | 8,575 | Graph | 588 | 1 |
| | tox21 | 18.6 | 38.6 | 7,831 | Graph | 12 | 1 |

## F.2 DATASET SPLIT

Our split protocol adheres to established standards to ensure reproducibility. Cora employs 10 predefined data partitions with varying random seeds, while WikiCS utilizes 20 distinct training splits each evaluated with 20 seed variations. For biochemical datasets HIV and MUV, we strictly follow their canonical test splits across 5 randomized trials. Knowledge graphs WN18RR and FB15K237 adopt the reference partitioning scheme from prior work, with all experiments repeated 5 times under different initialization conditions to compute stable performance metrics.

## F.3 FINE-TUNING AND FEW-SHOT EXPERIMENTAL IMPLEMENTATIONS

During fine-tuning, we follow [25] and leverage both prototype and linear classifiers. The prototype classifier constructs class-specific prototypes by averaging quantized embeddings for each category, then makes predictions through cosine similarity comparisons between embeddings and prototypes. In parallel, the linear classifier processes the same quantized embeddings through a trainable projection layer to generate predictions. Both classifiers are optimized using cross-entropy loss. During inference, we combine predictions from both classifiers to benefit from their complementary strengths.

The molecular graph classification framework accommodates diverse $n$-way binary classification scenarios. For instance, the HIV dataset is processed as a conventional binary classification task, while more complex datasets like MUV require multi-task binary classification across 128 distinct targets. In our few-shot learning implementation, we adopt an episodic training paradigm where each task consists of randomly sampled $k$-shot support sets and arbitrary unlabeled query instances.

Table 8: Additional performance on few-shot and zero-shot settings in Cora.

| Model | Cora - 7way | | | | Cora - 2way | | | |
|-------|-------------|---|---|---|-------------|---|---|---|
| | 10-shot | 5-shot | 3-shot | 0-shot | 10-shot | 5-shot | 3-shot | 0-shot |
| GraphMAE | $55.25_{\pm3.12}$ | $53.80_{\pm5.22}$ | $54.35_{\pm6.74}$ | $48.15_{\pm4.86}$ | $70.15_{\pm5.42}$ | $69.92_{\pm6.23}$ | $67.47_{\pm7.17}$ | $60.61_{\pm7.28}$ |
| GIANT | $56.43_{\pm3.63}$ | $55.28_{\pm4.87}$ | $54.86_{\pm6.28}$ | $50.65_{\pm5.52}$ | $70.93_{\pm4.95}$ | $68.64_{\pm7.76}$ | $66.78_{\pm6.65}$ | $61.82_{\pm6.77}$ |
| GFT | $63.27_{\pm4.48}$ | $59.42_{\pm5.65}$ | $57.33_{\pm5.84}$ | $51.12_{\pm6.03}$ | $76.16_{\pm4.73}$ | $75.95_{\pm7.32}$ | $72.38_{\pm8.18}$ | $66.57_{\pm6.26}$ |
| OFA | $62.15_{\pm3.37}$ | $57.23_{\pm4.42}$ | $55.27_{\pm6.59}$ | $52.36_{\pm5.78}$ | $72.24_{\pm4.51}$ | $73.18_{\pm6.15}$ | $70.76_{\pm7.92}$ | $63.53_{\pm5.97}$ |
| SAMGPT | $60.14_{\pm5.22}$ | $58.37_{\pm5.28}$ | $57.52_{\pm4.43}$ | $51.73_{\pm5.53}$ | $74.32_{\pm5.28}$ | $72.56_{\pm4.93}$ | $69.23_{\pm6.64}$ | $64.24_{\pm5.72}$ |
| UniGraph | $61.98_{\pm4.11}$ | $61.25_{\pm4.14}$ | $60.52_{\pm5.28}$ | $53.07_{\pm5.31}$ | $73.47_{\pm4.07}$ | $72.72_{\pm7.76}$ | $72.65_{\pm8.41}$ | $65.82_{\pm6.45}$ |
| **MoT** | $\mathbf{68.43}_{\pm4.58}$ | $\mathbf{65.24}_{\pm4.23}$ | $\mathbf{66.00}_{\pm5.46}$ | $\mathbf{61.67}_{\pm6.90}$ | $\mathbf{82.50}_{\pm5.98}$ | $\mathbf{82.33}_{\pm8.67}$ | $\mathbf{81.83}_{\pm9.26}$ | $\mathbf{68.00}_{\pm6.54}$ |

Table 9: Additional performance on few-shot and zero-shot settings in History.

| Model | History - 10way | | | | History - 2way | | | |
|---|---|---|---|---|---|---|---|---|
| | 10-shot | 5-shot | 3-shot | 0-shot | 10-shot | 5-shot | 3-shot | 0-shot |
| GraphMAE | $46.85_{\pm4.68}$ | $41.69_{\pm8.76}$ | $40.98_{\pm8.78}$ | $29.11_{\pm9.68}$ | $66.39_{\pm6.10}$ | $65.37_{\pm5.02}$ | $62.93_{\pm5.35}$ | $55.18_{\pm7.56}$ |
| GIANT | $46.51_{\pm4.29}$ | $43.95_{\pm9.22}$ | $38.91_{\pm9.26}$ | $29.73_{\pm9.03}$ | $67.55_{\pm5.68}$ | $65.28_{\pm5.95}$ | $63.63_{\pm6.94}$ | $56.78_{\pm7.16}$ |
| GFT | $56.18_{\pm6.30}$ | $50.29_{\pm9.49}$ | $50.21_{\pm8.32}$ | $35.09_{\pm6.72}$ | $73.29_{\pm4.97}$ | $\mathbf{72.30}_{\pm5.78}$ | $68.95_{\pm5.86}$ | $58.36_{\pm6.15}$ |
| OFA | $54.31_{\pm4.90}$ | $49.50_{\pm8.27}$ | $47.17_{\pm8.51}$ | $34.03_{\pm8.41}$ | $71.15_{\pm4.63}$ | $69.41_{\pm5.49}$ | $67.21_{\pm6.48}$ | $60.20_{\pm6.75}$ |
| SAMGPT | $56.04_{\pm6.78}$ | $49.82_{\pm8.44}$ | $48.02_{\pm9.06}$ | $36.43_{\pm8.56}$ | $72.20_{\pm5.27}$ | $70.33_{\pm5.59}$ | $66.78_{\pm6.08}$ | $61.57_{\pm5.94}$ |
| UniGraph | $54.75_{\pm7.65}$ | $52.40_{\pm8.57}$ | $48.94_{\pm8.29}$ | $34.94_{\pm7.02}$ | $74.91_{\pm4.62}$ | $70.38_{\pm5.53}$ | $69.53_{\pm6.40}$ | $60.93_{\pm7.67}$ |
| **MoT** | $\mathbf{57.60}_{\pm6.10}$ | $\mathbf{54.55}_{\pm7.23}$ | $\mathbf{51.28}_{\pm8.82}$ | $\mathbf{39.15}_{\pm7.66}$ | $\mathbf{75.16}_{\pm6.05}$ | $72.11_{\pm5.53}$ | $\mathbf{71.46}_{\pm6.71}$ | $\mathbf{64.21}_{\pm6.03}$ |

Table 10: Additional performance on few-shot and zero-shot settings in WN18RR.

| Model | WN18RR - 10way | | | | WN18RR - 2way | | | |
|---|---|---|---|---|---|---|---|---|
| | 10-shot | 5-shot | 3-shot | 0-shot | 10-shot | 5-shot | 3-shot | 0-shot |
| GraphMAE | $52.15_{\pm2.12}$ | $50.80_{\pm3.24}$ | $49.35_{\pm3.15}$ | $42.15_{\pm3.86}$ | $76.15_{\pm5.42}$ | $74.92_{\pm6.23}$ | $73.47_{\pm6.17}$ | $54.61_{\pm7.28}$ |
| GIANT | $53.43_{\pm3.63}$ | $51.28_{\pm2.73}$ | $50.86_{\pm4.28}$ | $43.65_{\pm5.52}$ | $78.93_{\pm4.95}$ | $76.64_{\pm6.76}$ | $72.78_{\pm7.65}$ | $55.82_{\pm6.77}$ |
| GFT | $55.27_{\pm3.48}$ | $53.42_{\pm2.13}$ | $53.33_{\pm3.84}$ | $45.12_{\pm4.03}$ | $80.16_{\pm6.73}$ | $77.95_{\pm7.32}$ | $75.38_{\pm5.18}$ | $\mathbf{62.57}_{\pm8.26}$ |
| OFA | $55.15_{\pm4.37}$ | $52.23_{\pm3.42}$ | $52.27_{\pm5.59}$ | $44.36_{\pm5.78}$ | $81.24_{\pm6.51}$ | $77.18_{\pm7.15}$ | $74.76_{\pm6.92}$ | $59.53_{\pm9.97}$ |
| SAMGPT | $53.14_{\pm4.22}$ | $51.37_{\pm3.28}$ | $50.52_{\pm4.43}$ | $45.73_{\pm5.53}$ | $80.32_{\pm6.28}$ | $78.56_{\pm8.93}$ | $75.23_{\pm7.64}$ | $58.24_{\pm8.72}$ |
| UniGraph | $56.98_{\pm2.11}$ | $55.25_{\pm3.14}$ | $54.52_{\pm3.28}$ | $47.07_{\pm5.31}$ | $82.47_{\pm5.07}$ | $81.72_{\pm7.76}$ | $77.65_{\pm7.41}$ | $60.82_{\pm9.45}$ |
| **MoT** | $\mathbf{64.07}_{\pm3.64}$ | $\mathbf{62.13}_{\pm3.26}$ | $\mathbf{60.80}_{\pm4.35}$ | $\mathbf{50.13}_{\pm5.77}$ | $\mathbf{84.00}_{\pm6.72}$ | $\mathbf{82.67}_{\pm8.98}$ | $\mathbf{82.00}_{\pm6.94}$ | $61.33_{\pm9.15}$ |

## G ADDITIONAL EXPERIMENT RESULTS

We present more extensive few-shot and zero-shot experiments in Table 8, 9 and 10, which reveal consistent patterns. As classification complexity increases with higher $n$-way configurations, all models exhibit performance degradation due to expanding decision boundaries. Similarly, reducing support samples ($k$-shot) amplifies performance decay as limited supervision fails to capture class distinctions. Crucially, MoT consistently outperforms all baselines across these challenging scenarios. This robustness validates MoT's effectiveness in preserving semantic separability despite increasing task difficulty and decreasing supervision.

Table 11: Fine-tuning hyperparameter configurations across datasets.

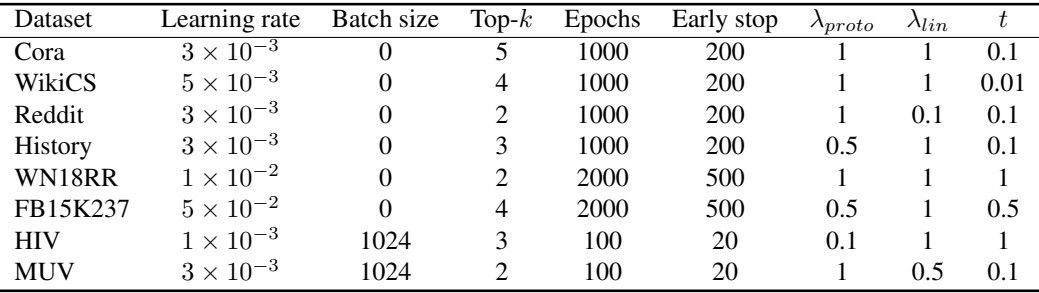

| Dataset | Learning rate | Batch size | Top-$k$ | Epochs | Early stop | $\lambda_{proto}$ | $\lambda_{lin}$ | $t$ |
|---|---|---|---|---|---|---|---|---|
| Cora | $3 \times 10^{-3}$ | 0 | 5 | 1000 | 200 | 1 | 1 | 0.1 |
| WikiCS | $5 \times 10^{-3}$ | 0 | 4 | 1000 | 200 | 1 | 1 | 0.01 |
| Reddit | $3 \times 10^{-3}$ | 0 | 2 | 1000 | 200 | 1 | 0.1 | 0.1 |
| History | $3 \times 10^{-3}$ | 0 | 3 | 1000 | 200 | 0.5 | 1 | 0.1 |
| WN18RR | $1 \times 10^{-2}$ | 0 | 2 | 2000 | 500 | 1 | 1 | 1 |
| FB15K237 | $5 \times 10^{-2}$ | 0 | 4 | 2000 | 500 | 0.5 | 1 | 0.5 |
| HIV | $1 \times 10^{-3}$ | 1024 | 3 | 100 | 20 | 0.1 | 1 | 1 |
| MUV | $3 \times 10^{-3}$ | 1024 | 2 | 100 | 20 | 1 | 0.5 | 0.1 |

## H HYPER-PARAMETERS SETTING

The pre-training process employs a carefully designed set of hyperparameters to optimize the self-supervised learning objective across diverse graph datasets. We utilize the AdamW optimizer with a learning rate of $1 \times 10^{-4}$ and weight decay of $1 \times 10^{-5}$ for 5 training epochs with batch size 1024. The model architecture consists of a 2-layer graph encoder with hidden dimension 768 and dropout rate 0.15, using ReLU activation functions and batch normalization. The masking strategy employs feature and edge masking probabilities 0.1, while the loss function combines feature reconstruction ($\lambda_1 = 100$), topology reconstruction ($\lambda_2 = 0.01$), contrastive learning ($\lambda_3 = 0.001$), and domain alignment ($\lambda_4 = 0.01$) components. All experiments use random seed 42 for reproducibility.

Fine-tuning hyperparameters are specifically optimized for each dataset through extensive grid search, with key configurations summarized in Table 11. The two loss coefficients $\lambda_{proto}$ and $\lambda_{lin}$ represent the relative weights of the prototype classifier and linear classifier losses during fine-tuning, respectively. During inference, the final prediction is obtained by combining outputs from both classifiers through a weighted fusion mechanism, where the trade-off parameter $t$ determines the contribution weight of the linear classifier ($t$) and the prototype classifier ($1 - t$).

