# OpenReview forum: "Two Sides of the Same Optimization Coin: Model Degradation and Representation Collapse in Graph Foundation Models"
_ICLR.cc/2026/Conference — ICLR 2026 Conference Withdrawn Submission_

### Official Review · Reviewer_35LW · 2025-10-16

**Soundness:** 2
**Presentation:** 1
**Contribution:** 2
**Rating:** 2
**Confidence:** 5

**Summary:**

This paper studies graph foundation model, and proposes a MoE module together with a graph self-supervised learning-based regularization term to enhance the performance.

**Strengths:**

1. The general layout of the paper is good
2. the performance improvement is ok

**Weaknesses:**

In general, I think the direction studied in this paper is not very promising, and the paper's writing is very confusing.

1. In the abstract, the definition of GFM is problematic. Why does GFM have to be working on text-attributed graphs? Many GFMs are working beyond TAGs, like [1,2].
2. Similarly, in the introduction part, the author directly goes to gVQ-MAEs? What is gVQ-MAE? I know there are some works like GFT that use the VAE as "transferable vocabulary tokens". This is just a small portion of GFM research, and I don't see any of these works presenting promising results. For node classification tasks, I remember GFT can't even beat the GNN baselines trained from scratch. So, I doubt the value of studying this line of work.
3. What's the pitfall of gVQ-MAEs? I can only see Figure 2. You should first use one section to introduce these phenomena and then go to the latter method sections.
4. For line 132,  you should cite relevant papers using token IDs in GFMs instead of citing this paper using token Ids for knowledge distillation.
5. In Line 144, I'm very confused about what's "model degradation"? You should first have a preliminary section talking about these two problems.
6. The appearance of Section 3 is very confusing. Why we come up with these questions? You should first state the motivation.
7. the theoretrical part is some very general math formulation which doesn't even touch the graph structure. It doesn't help motivate your proposed methods
8. In terms of KG, is link classification a reasonable task? Usually we deal with knowledge completion in KG, in which you need to compare pairwise information with all candidates.



[1] Galkin M, Yuan X, Mostafa H, et al. Towards foundation models for knowledge graph reasoning[J]. arXiv preprint arXiv:2310.04562, 2023.
[2] Méndez-Lucio, O., Nicolaou, C.A. & Earnshaw, B. MolE: a foundation model for molecular graphs using disentangled attention. Nat Commun 15, 9431 (2024). https://doi.org/10.1038/s41467-024-53751-y
[3] Wang Z, Zhang Z, Chawla N, et al. Gft: Graph foundation model with transferable tree vocabulary[J]. Advances in Neural Information Processing Systems, 2024, 37: 107403-107443.

**Questions:**

What's the merit of using vector quantization in the graph foundation model? As from [1], I acknowledge that these kinds of IDs can enhance efficiency. Based on my experience with cross-domain tasks like recommendation (which uses semantic IDs), I believe the common conclusion is that these IDs are not transferable.

[1] Luo, Yuankai, et al. "Node identifiers: Compact, discrete representations for efficient graph learning." arXiv preprint arXiv:2405.16435 (2024).

---

### Official Review · Reviewer_QYw5 · 2025-10-20

**Soundness:** 1
**Presentation:** 1
**Contribution:** 2
**Rating:** 2
**Confidence:** 5

**Summary:**

The paper studies graph foundation models (GFM) through the lens of graph oriented GFMs. The authors first identifies current bottlenecks in training gVQ-MAEs and propose several tricks to improve the optimization and regularization of the training process. Empirical results are competitive.

**Strengths:**

The intersection of graph representation learning and foundation models such as LLMs is a promising area of research.

**Weaknesses:**

First of all, I would like to point out a critical flaw in this paper.

## The theoretical statements are wrong in a FUNDAMENTAL SENSE

Upon primary investigations of the theory statements in section 4.3 as well as proofs provided in appendix E, I found the proofs to be fundamentally flawed. Specifically:
- In theorem 4.2, the intuition is correct since MoT incorporates a larger function class than that of vanilla edge representation construction via allowing node-to-edge dual message passing. However, upon checking the proof it seems that equation 15 in appendix E is NOT a valid lower bound, for example take $q(\mathcal{E} | Z) \equiv 100$ as a constant encoder, then we simply arrive at $I(Z, \mathcal{E}) \ge \log 100 + H(\mathcal{E})$ which violates the fundamental inequality in information theory that mutual information is upper bounded by entropy. Additionally, I checked the reference [1] and there seems to contain no such variational lower bounds of MI (lower bounding MI is non-trivial and often obtained lower bounds are not very meaningful). If the authors did find a place that validates this lower bound please state it in a clear fashion, **I think this bound is incorrect, and therefore the MI gap proof is likely to be incorrect as well**------it seems that the logic in the proof thereafter are somewhat messy.
- In theorem 4.3, the statement itself is problematic: As there are altogether $M$ codebooks with each containing $K$ codewords, the entropy of any distribution supported on this collection is upper bounded by $\log(MK)$ which further upper bounds MI. It is therefore strange to see the authors using $\log(MK)$ as an MI lower bound. Upon investigation of the proofs in section E.2, once again it seems everywhere contains mistakes------In equation 23, what does *the probability of correct domain-specific mapping* mean? it appears pretty awkward to me that a uniform probability somehow emerges as some correct choice without formal justification. A more critical error comes right after in equation 25 that uses the inequality $I(S; Z) \ge H(S) - H(S|Z)$, but **this is a well known equality that defines MI**. [2]

That said, I think the two aforementioned errors are rather critical, as even researchers with entry-level knowledge of information theory would identify the incorrectness of such statements, which makes me seriously suspect that LLMs are abused in writing such hallucinatory statements without careful fact checking. Furthermore, I noticed that there is no disclosure of LLM usage segments in the appendix of this paper.

## Novelty issues
The technique of edge-wise semantic fusion described in equation 2 is actually a well-known technique in graph representation learning community that utilizes the edge-to-node duality, or line graph in the undirected graph case. Such adaptation are present even in pioneer works of GNN developments [1] and has later turned into broader protocols [2, 3]. Therefore I do not think this idea is new.

## Presentation issues
While there are plenty of summaries appeared at the start of paragraphs, I think the contents of the paper are organized in a non-natural way. For example, in section 3.2 the authors presented empirical investigation with improvements brought by the proposed methods, but without description of the method. I think perhaps this is a little bit harsh.

**Questions:**

The most important question is: **How did you get your theory results?**

[1]. Alemi, Alexander A., et al. "Deep variational information bottleneck." arXiv preprint arXiv:1612.00410 (2016).
[2]. Cover, Thomas M. Elements of information theory. John Wiley & Sons, 1999.
[3]. Chen, Zhengdao, Lisha Li, and Joan Bruna. "Supervised Community Detection with Line Graph Neural Networks." International conference on learning representations. 2020.
[4]. Jo, Jaehyeong, et al. "Edge representation learning with hypergraphs." Advances in Neural Information Processing Systems 34 (2021): 7534-7546.
[5]. Wu, Ruofan, et al. "Grande: a neural model over directed multigraphs with application to anti-money laundering." 2022 IEEE International Conference on Data Mining (ICDM). IEEE, 2022.

---

### Official Review · Reviewer_swej · 2025-10-30

**Soundness:** 3
**Presentation:** 4
**Contribution:** 3
**Rating:** 6
**Confidence:** 4

**Summary:**

This paper investigates critical optimization challenges in graph foundation models (GFMs), particularly focusing on graph vector-quantized masked autoencoders. It identifies two interrelated pitfalls that arise during multi-domain pre-training: failure to capture input diversity and loss of semantic separability. They propose Mixture-of-Tinkers (MoT), which integrates an Information Tinker (edge-wise semantic fusion and mixture-of-codebooks) and a Regularization Tinker (contrastive alignment and load-balancing constraint). In addition, the study provides theoretical analyses that connect MoT with the Information Bottleneck principle. The extensive experiments on 22 datasets across 6 domains demonstrate state-of-the-art results under supervised, few-shot and zero-shot settings.

**Strengths:**

1. The paper presents preliminary experiments that reveal inherent issues concerning model degradation and representation collapse in existing GFMs.
2. The proposed MoT framework provides clear mitigation strategies including information and regularization tinkers. The study provides a theoretical analysis for the Mixture-of-Tinkers from the perspective of the Information Bottleneck principle.
3. The paper provides a broad range of empirical evidence demonstrating that MoT outperforms existing GFM baselines across multiple domains and training paradigms.

**Weaknesses:**

1. While MoT achieves clear performance gains, the proposed framework is an incremental improvement built upon VQ-VAE and MoE variants. The two regularization objectives are closely related to the commitment loss and expert-load loss used in prior works.

2. The adaptation of the Information Bottleneck (IB) in Definition 4.1 appears inconsistent with prior IB methods [1][2]. Instead of constraining the latent space to retain only minimal sufficient information within the input graph data, the objective in Definition 4.1 simultaneously maximizes mutual information with domain-specific semantics $S$ and graph topology $\mathcal{E}$. It appears to overemphasize domain-dependent semantic/topological information and weaken the theoretical connection to the proposed framework.

3. Most experiments focus solely on the accuracy. The paper lacks deeper analyses of representation quality (e.g., the similar experimental results in Fig. 2) to substantiate its claim of mitigating representation collapse.


[1] Graph Information Bottleneck, 2020.

[2] Deep learning and the information bottleneck principle, 2015.

**Questions:**

1. How is the *domain generalization conflict* formally defined in this work? Furthermore, what distinguishes the proposed Information Bottleneck formulation for GFMs from conventional graph IB methods? It would be helpful to include additional analyses or visualizations (e.g., KL Divergence Heatmap in Fig. 2) to validate the claimed theoretical connection.

2. Introducing the Mixture-of-Codebooks (MoC) may cause additional challenges of load imbalance or codebook collapse. Beyond accuracy-based metrics,  other metrics (e.g., codebook utilization and expert activation) are necessary for a more insightful analysis of the MoC.

3. The functionality of the MoC appears achievable through simpler and more established vector quantization methods. Could the proposed Mixture-of-Codebooks be simplified or replaced with a more computationally efficient quantization variant (e.g., RQ-VAE or OPQ) in the ablation study?

---

### Official Review · Reviewer_Ap7s · 2025-10-31

**Soundness:** 1
**Presentation:** 1
**Contribution:** 1
**Rating:** 2
**Confidence:** 2

**Summary:**

I am unable to recommend acceptance for this paper. My primary reason is that the paper, in its current form, is not reviewable due to severe issues with clarity, structure, and presentation. While the proposed method may have technical merit, the current manuscript is unintelligible, making it impossible to assess the validity of the approach or the significance of the results.

My core concerns are centered on the quality of the writing and presentation:
- The paper fails to provide sufficient context for the problem being addressed. Section 3, in particular, introduces experiments without a clear explanation of their purpose, the hypotheses being tested, or how they connect to the paper's broader claims. The reader is left to guess the motivation behind the experimental design.
- The figures and diagrams provided are overly dense and they do not serve as an effective aid for understanding. The captions are minimal and fail to explain the figures or what the reader should conclude from them. A good figure should be largely understandable on its own, and these are not.
- The paper's structure does not follow a logical narrative. Most paragraphs, instead of building an argument, are a dense collection of bullet points. This style of writing makes it incredibly difficult to follow the authors' reasoning or understand the flow of the proposed method.

It is entirely possible that the research described in this paper is solid and contains valuable contributions. However, in the absence of an intelligible presentation, I cannot provide a technical assessment.

**Strengths:**

Look at summary

**Weaknesses:**

Look at summary

**Questions:**

Look at summary

---

### Note · Authors · 2025-11-16

I have read and agree with the venue's withdrawal policy on behalf of myself and my co-authors.